# The Assessment of the Phototoxic Action of Chlortetracycline and Doxycycline as a Potential Treatment of Melanotic Melanoma—Biochemical and Molecular Studies on COLO 829 and G-361 Cell Lines

**DOI:** 10.3390/ijms24032353

**Published:** 2023-01-25

**Authors:** Jakub Rok, Zuzanna Rzepka, Klaudia Banach, Justyna Kowalska, Dorota Wrześniok

**Affiliations:** Department of Pharmaceutical Chemistry, Faculty of Pharmaceutical Sciences in Sosnowiec, Medical University of Silesia, Jagiellońska 4, 41-200 Sosnowiec, Poland

**Keywords:** doxycycline, chlortetracycline, phototoxicity, UVA radiation, melanoma, apoptosis

## Abstract

Melanoma is still one of the most dangerous cancers. New methods of treatment are sought due to its high aggressiveness and the relatively low effectiveness of therapies. Tetracyclines are drugs exhibiting anticancer activity. Previous studies have also shown their activity against melanoma cells. The possibility of tetracycline accumulation in pigmented tissues and the increase in their toxicity under the influence of UVA radiation creates the possibility of developing a new anti-melanoma therapy. This study aimed to analyze the phototoxic effect of doxycycline and chlortetracycline on melanotic melanoma cells COLO 829 and G-361. The results indicated that tetracycline-induced phototoxicity significantly decreased the number of live cells by cell cycle arrest as well as a decrease in cell viability. The simultaneous exposure of cells to drugs and UVA caused the depolarization of mitochondria as well as inducing oxidative stress and apoptosis. It was found that the combined treatment activated initiator and effector caspases, caused DNA fragmentation and elevated p53 level. Finally, it was concluded that doxycycline demonstrated a stronger cytotoxic and phototoxic effect. UVA irradiation of melanoma cells treated with doxycycline and chlortetracycline allows for the reduction of therapeutic drug concentrations and increases the effectiveness of tested tetracyclines.

## 1. Introduction

Melanoma develops as a result of the neoplastic transformation of melanocytes. Pathological changes include genetic, epigenetic, and allogeneic modifications [1]. Despite the growing knowledge of patients about melanoma, including symptoms and risk factors, epidemiological data are still disturbing.

Although melanoma is not the most common neoplastic disease of the skin, it is responsible for the most deaths caused by skin cancers [2,3,4,5]. Over the past few decades, the number of newly diagnosed cases of this cancer has increased significantly. It was found that melanoma is characteristic of middle-aged patients (45–50 years). Nevertheless, it is also the second most common cancer among all cancers diagnosed in young people aged 25–29 years [6,7]. Currently applied therapeutic methods to treat melanoma skin cancer include, among others, surgical resection, chemotherapy and photodynamic therapy. In addition, targeted therapy and immunotherapy have been introduced in recent years. Although these methods have resulted in the extension of patients’ lives, new and better therapies with better safety and higher clinical effectiveness are still being sought [8,9].

Tetracyclines are broad-spectrum polyketide antibiotics that exert a bacteriostatic effect by reversible binding to the bacterial 30S ribosomal subunit and blocking incoming aminoacyl tRNA from binding to the ribosome acceptor site [10,11,12]. Doxycycline, a second-generation tetracycline, is one of the most commonly prescribed antibiotics for patients with acne, urinary tract infections, sexually transmitted diseases, and abdominal aortic aneurysms [13,14]. In turn, chlortetracycline is a first-generation tetracycline. This antibiotic is used mainly in dermatology and veterinary medicine [15,16]. In recent years, researchers have been focusing on analyzing and studying the anticancer properties of tetracyclines. It was found that the drugs inhibited the migration and adhesion of cancer cells. Moreover, doxycycline affected cell growth and proliferation as well as inducing apoptosis. However, molecular mechanisms of the antitumor action of tetracyclines have not been fully understood [17,18,19].

Skin adverse effects are among the characteristics of tetracyclines. It was found that the drugs exhibited the ability to form stable complexes with melanin, a pigment produced by melanocytes. The binding of tetracyclines to melanin resulted in their retention and accumulation in pigmented tissues, such as the skin. The presence of drugs in high concentrations in melanin-containing cells may contribute to an increase in their cytotoxic effects [20,21,22]. Previous studies have indicated that tetracyclines decreased the viability of normal human melanocytes. Their cytotoxic potential against melanocytes rises in the following order: chlortetracycline < tetracycline < doxycycline [23,24,25]. Considering tetracycline accumulation in the skin, and their cytotoxic and anticancer potential, it is reasonable to investigate the anti-melanoma properties of the drugs [26].

It is well known that tetracyclines may cause cutaneous phototoxic reactions. Clinical manifestations include edema, erythema, urticaria, blisters, and photo-onycholysis. The phototoxic properties of tetracyclines result from their broad absorption spectrum in the UVA range [27]. The tetracycline molecule easily absorbs the radiation, resulting in the transition of the molecule from the ground state to the excited single state. The excited molecule returns to the ground state, e.g., by transferring energy to the oxygen molecule. This process is directly associated with the production of reactive oxygen species (ROS). In addition, under UVA radiation and ROS, tetracyclines are converted to toxic photoproducts. Both photoproducts and ROS are harmful and lead to damage to skin cells [28,29]. Various tetracyclines differ in their phototoxic activity. Studies on normal human melanocytes demonstrated that the phototoxic potential of tetracyclines increased in the following order: doxycycline < tetracycline < chlortetracycline [23,24,25].

Considering the above issues, the possibility of using the phototoxic effect of doxycycline and chlortetracycline against melanotic melanoma cells was investigated.

## 2. Results

### 2.1. The Cytotoxic and Phototoxic Actions of Doxycycline and Chlortetracycline Cause a Decrease in the Percentage of Metabolically Active Melanoma Cells and Disturb Cell Cycle

A preliminary assessment of cytotoxicity and phototoxicity of the tested tetracyclines was performed using the WST-1 assay (Figure 1A–D). Both melanotic melanoma cell lines were analyzed 24 h after the irradiation with UVA. Before the exposure to UVA, the cells were incubated with doxycycline or chlortetracycline in concentrations ranging from 10 µM to 200 µM. The obtained results showed that both drugs decreased the percentage of metabolically activated cells proportionally to the concentration. The exposure of treated cells to UVA radiation augmented the effect significantly. The obtained EC_50_ values (Table 1) indicated that doxycycline was more cytotoxic than chlortetracycline in both cell lines. The EC_50_ parameters for chlortetracycline were over 3.6 times higher than those calculated for doxycycline. At the same time, it should be noted that individual melanoma lines showed different sensitivity to the cytotoxic effects of the drugs. Moreover, the G-361 cell line appeared to be more amenable to the treatment with tetracyclines. The EC_50_ values for doxycycline and chlortetracycline were more than two times lower for the G361 cells when compared to COLO 829. The lowest EC_50_ values were noted for G-361 cells treated with doxycycline: 23.3 µM and 11.5 µM for non-irradiated and irradiated cultures, respectively. On the other hand, the highest values were observed for chlortetracycline and COLO 829 cells: 185.2 µM and 52.0 µM for non-exposed and exposed to UVA cultures, respectively.

Based on the obtained EC_50_ values, the photo-irritation factor (PIF) coefficient was also calculated (Table 1). PIF expresses the ratio of drug toxicity in non-irradiated samples to toxicity in samples exposed to radiation. Thus, the coefficient allows the assessment of the impact of UVA radiation on the increase in cytotoxicity, and in this case, also the assessment of the effectiveness of anti-melanoma therapy based on tetracycline phototoxicity. The calculated values of PIF suggested that UVA radiation significantly enhanced the cytotoxic effect of DOX and CHL on melanoma cells. The phototoxic potential was similar for both tested tetracyclines, but it was cell line-dependent. It was noticed that PIF values were higher for COLO 829 cells, which suggested greater sensitivity to phototoxic effects.

Taking into account the presented results of the WST-1 test, doxycycline at a concentration of 50 µM and chlortetracycline at 150 µM were chosen in subsequent studies.

One of the possible reasons for the observed decrease in the percentage of metabolically active cells was the antiproliferative action of the tested tetracyclines. For this reason, the cell cycle was analyzed. The obtained results were presented as the percentage of cells in individual phases (Figure 1E,F) as well as relative ratios of G_1_/S and G_2_-M/S (Table 2) which characterized a type of cell cycle arrest. Based on the analysis, it was found that the cell cycle in tested melanoma cultures was dysregulated after the treatment. Overall, the exposure of the melanoma cells to UVA radiation alone resulted in slight changes in the cell cycle profile. On the other hand, the incubation of cells with doxycycline significantly increased the value of G_1_/S ratio for both cell lines and the G_2_-M/S ratio for COLO 829 cells. In turn, CHL caused only a significant increase in G_1_/S ratio for COLO 829. Simultaneous exposure of cell cultures to CHL and UVA was additionally associated with an increase in the values of the G_2_-M/S ratio for COLO 829 and the G_1_/S ratio for G-361 cells. For melanoma cells simultaneously exposed to DOX and UVA, there was a significant decrease in the G_1_/S ratio for COLO 829 and the G_2_-M/S ratio for G-361 cells. Regardless of the analysis of the values, it should be noted that doxycycline in irradiated cells caused a very significant decrease in the percentage of cells in the G_1_ phase (35% vs. 65% in the control for COLO 829 cells; 24% vs. 60% in the control for G-361 cells). Such an effect was not reported for chlortetracycline. It should also be emphasized that the phototoxic effect of the tetracyclines increased the percentage of cells in the sub-G_1_ phase. This effect was noted for doxycycline in both cell types (23% for COLO 829 and 67% for G-361) and chlortetracycline in the G-361 cells (22%).

### 2.2. Phototoxic Action of Doxycycline and Chlortetracycline Causes a Decrease in Melanoma Cell Viability

Cell viability was analyzed using imaging cytometry. The obtained results for individual samples and the corresponding microscopic images are shown in Figure 2. Based on this study, it was found that the viability of cells exposed only to doxycycline at a concentration of 50 µM or chlortetracycline at 150 µM was high and was not different from controls. Additionally, the applied dose of UVA radiation did not affect melanoma cell viability. A significant increase in the percentage of dead cells was observed in cultures exposed to both drug and UVA. The greatest decrease in viability was noted in cultures incubated with doxycycline and irradiated. In this case, the percentage of live cells was 27.3% for COLO 829 and 31.8% for G-361. The corresponding results for chlortetracycline were 87.7% and 65.1%, respectively.

The changes in cell viability caused by the phototoxicity of tetracyclines were reflected in the microscopic images of cell cultures (Figure 2B,D). The cell abundance and morphology were observed using the light inverted microscope NIKON TS100F (Nikon Instruments, Amsterdam, The Netherlands). The images showed not only a significant decrease in the number of cells but also differences in the morphology of melanoma cells. In general, the cell cultures exposed to only one of the tested drugs were characterized mainly by a reduction in the cell number. In cultures additionally irradiated with UVA, it was noted that cells were smaller, rounded and separated than in the control. The loss of adherence, intercellular contact as well as a characteristic shape suggested that the phototoxic action of tetracyclines induced apoptosis.

### 2.3. Analysis of Apoptosis in Melanoma Cells Irradiated with UVA and Treated with Doxycycline and Chlortetracycline

The assessment of apoptosis was made using cytometric analysis of annexin V—a molecular marker of programmed cell death. The obtained results are presented as bar graphs and representative scatter plots (Figure 3). It was noticed that both tested tetracyclines were able to induce apoptosis in COLO 829 and G-361 melanoma cells. A stronger effect was found for doxycycline. The percentage of annexin V-positive cells for DOX was about 47% in COLO 829 and G-361 cells. In turn, chlortetracycline induced apoptosis in about 25% and 27% for COLO 829 and G-361, respectively. The exposure of cells to UVA radiation enhanced the effect significantly for both drugs. The highest percentage of apoptotic cells was found for DOX and UVA-exposed cultures: 94% for COLO 829 and 89% for G-361. The observed effect was weaker for CHL. The corresponding results were 71% for COLO 829 and 75% for G-361. At the same time, it should be added that UVA radiation alone did not significantly affect the induction of the apoptosis process.

### 2.4. Analysis of Mitochondrial Membrane Potential (ψm) in Melanoma Cells Treated with Doxycycline or Chlortetracycline and Irradiated with UVA

The influence of the cytotoxic and phototoxic action of DOX and CHL on mitochondrial membrane potential (MMP) was measured cytometrically. The obtained results are presented in Figure 4 as bar graphs and representative scatter plots. The analysis indicated that the melanoma cells treated with CHL or DOX were characterized by a decreased mitochondrial membrane potential. Stronger action was observed for G-361 melanoma cells. In this case, the percentage of cells with a depolarized mitochondrial membrane was about 40% and 54% for chlortetracycline and doxycycline, respectively. The exposure of cells to UVA radiation enhanced the effect of drugs, regardless of the cell line. Of note, a greater increase in the population of cells with reduced MMP was found for COLO 829. The increase was from 17% to 66% for CHL and from 21% to 46% for DOX. The corresponding results for the G-361 cell line were from 40% to 55% and from 54% to 64%, respectively.

### 2.5. The Assessment of Redox Homeostasis in Melanoma Cells Treated with Doxycycline and Chlortetracycline and Exposed to UVA Radiation

Redox homeostasis in the tested cells was evaluated by the assessment of the intracellular level of reduced and oxidized glutathione (GSH) as well as by the measurement of the relative level of ROS. The obtained results are presented in Figure 5. It was found that the treatment only with the selected tetracyclines did not affect intracellular thiols in melanoma cells. A slight change in the GSH ratio was observed after irradiation of COLO 829 and G-361 cells with UVA. Very large decreases in the ratio were caused by the simultaneous exposure of the tested cells to the drug and UVA radiation, regardless of the cell line. A decrease in the value of the ratio means an increase in the percentage of cells with a high level of the oxidized form of glutathione. The phototoxic effect of doxycycline was found to be stronger compared to the phototoxic effect of chlortetracycline. When doxycycline and UVA were used, low content of reduced thiols was found in over 80% and 90% of COLO 829 and G-361 cells, respectively.

The results of thiol analysis were reflected in the results for intracellular ROS levels. The highest values for the ROS analysis were observed in cells simultaneously exposed to UVA and doxycycline: 302% in COLO 829 and 418% in G-361 cells. The corresponding results for treatment with chlortetracycline and UVA were 175% and 248%, respectively. The level of ROS in cells exposed to UVA alone was not changed in both cell lines significantly. An increase in ROS production in melanoma cells treated with one of the tested tetracyclines was noticed only for doxycycline and G-361 cells. In this case, the result was 149%.

### 2.6. Confocal Imaging of Melanotic Melanoma Cells Treated with Doxycycline and Chlortetracycline and Exposed to UVA Radiation

All samples imaged using confocal microscopy were fixed 24 h after the irradiation. Cell staining allowed visualization of actin filaments, nuclei as well as the p53 protein. Representative pictures are presented in Figure 6 as merged images as well as separated images showing the green channel for p53. It was observed that cells exposed only to UVA radiation did not differ from control samples. In the remaining samples, a decrease in the number of cells and morphological changes were noticed. The changes included the loosening of the cell colony structure and the occurrence of singly growing cells. The observations indicated that both tetracyclines increased the level of p53 in melanoma cells and additional exposure to UVA enhanced the effect. A relatively high expression of p53 was found in COLO 829 cells after the simultaneous drug treatment and irradiation with UVA. A similar result for G-361 cells was noted only for samples exposed to chlortetracycline and UVA radiation.

### 2.7. The Influence of Doxycycline and Chlortetracycline on Caspase Activation and DNA Fragmentation in Melanotic Melanoma Cells Exposed to UVA Radiation

The activation of caspases and DNA fragmentation belong to major markers of serious cell damage and initiated the apoptosis process (Figure 7). The results obtained for COLO 829 cells indicated that the phototoxic action of DOX is significantly stronger than observed for CHL. The active caspases 3/7, 8, and 9 were found in 87%, 93%, and 96% of COLO 829 cells after simultaneous exposure to doxycycline and UVA, respectively. The corresponding results for CHL were 22%, 38%, and 11%, respectively. A greater effect of chlortetracycline phototoxicity on caspase activation was observed in G-361 melanoma cells. In this case, the percentage of cells with activated caspases was 44%, 64%, and 61% for caspases 3/7, 8, and 9, respectively. It should be noted that after UVA irradiation doxycycline activated all caspases in over 90% of G-361 cells.

Doxycycline caused DNA fragmentation in irradiated melanoma cells to the greatest extent compared to chlortetracycline. The percentage of cells with fragmented DNA after the treatment was 38% and 79% for COLO 829 and G-361, respectively. In turn, the exposure of cells to chlortetracycline and UVA radiation significantly increased the number of cells with fragmented DNA only for the G-361 cell line. In this case, DNA fragmentation was observed in 39% of the cell population.

## 3. Discussion

New anticancer therapies are still being sought. One of the main purposes of the development of such therapies is to improve the effectiveness and safety of treatment methods. Systemic use of drugs with high cytotoxic potential is usually associated with frequent and serious adverse reactions. One of the possibilities to reduce harmful side effects is to try to restrict the cellular damage to the tumor location. This possibility is provided using a combination of photoactive substances and precise irradiation with the appropriate wavelength. Currently, such a solution is applied in photodynamic therapy and photochemotherapy. The term “photodynamic action” was introduced by Hermann Von Tappeiner and Albert Jodbauer in 1907 based on previously conducted studies on skin tumors treated with eosin and exposed to white light [30]. It is worth emphasizing that photodynamic therapy was the first drug/device combination approved by the Food and Drug Administration [31]. The mechanism of PDT is a transfer of absorbed energy from a photo-excited molecule to molecular oxygen which generates ROS. In turn, the mechanism of photochemotherapy is oxygen-independent and includes other photochemical reactions, e.g., photoaddition to DNA [30]. Both above-mentioned therapies are used mainly in dermatology (e.g., in psoriasis and vitiligo) as well as in oncology (e.g., bladder, lung, or skin cancers) [30,31,32].

Tetracyclines belong to photo-excited molecules under UVA radiation. This property is the cause of skin adverse effects such as sunburn, blisters, and discoloration of the skin. However, there is a possibility that the observed phototoxic effect can be used for therapeutic purposes, e.g., to damage cancer cells. In this study, the phototoxic properties of tetracyclines were investigated in melanotic melanoma cells COLO 829 and G-361. Two tetracyclines, chlortetracycline and doxycycline, were selected for the analyses. Both drugs can form complexes with melanin which may lead to tetracycline accumulation in melanin-containing cells [20,21,23]. Moreover, the basis for the choice was also the phototoxic properties and the results of previous studies on skin cells. Although all tetracyclines have a similar structure, the comparative study revealed that the drugs differed in the strength of the phototoxic effect. It was found that doxycycline and chlortetracycline had a markedly greater phototoxic potential to keratinocytes than other drugs [33]. In turn, the study on darkly pigmented normal human melanocytes showed that doxycycline was significantly more cytotoxic than chlortetracycline but had a lower phototoxic potential. The EC_50_ values were 40 µM for DOX and 250 µM for CHL. In turn, the cell viability decreased after simultaneous exposure to UVA radiation and the drug in the EC_50_ concentration to 35.2% for DOX and 7.8% for CHL [24,25].

This study indicated that doxycycline and chlortetracycline showed cytotoxic and phototoxic effects against melanoma cells. However, the drugs differed in their potential. It was found that both actions of doxycycline appeared to be more powerful regardless of the tested cell line. The obtained values of EC_50_ for DOX were lower and the calculated PIF was greater than for CHL. Nevertheless, the effect of the drugs on normal skin cells should be also considered in the final evaluation. Taking into account the results obtained during the study on melanocytes, chlortetracycline seems to be a drug with a greater safety buffer. EC_50_ values for chlortetracycline were lower for melanoma cells than for normal melanocytes. In the case of doxycycline, such an effect was observed only for the G-361 cell line. The concentration EC_50_ of DOX for the COLO 829 cell line was slightly higher than for melanocytes. Nevertheless, assuming the precise photoactivation of the drug by irradiation, both drugs showed a therapeutic potential. The exposure of drug-treated cells resulted in an increase in toxicity of approximately two-fold for chlortetracycline and more than three-fold for doxycycline.

The observed decrease in melanoma cell number after the treatment was caused by the cell cycle arrest as well as by an increase in cell death. The changes in the percentage of cells in individual phases of the cell cycle were not specific for cytotoxic or phototoxic action. This suggests that the disturbance of the cell cycle is rather an effect of cellular damage than a direct and specific mechanism of action.

One of the possible reasons for the noted changes in the cell cycle profile was the induction of oxidative stress. The condition appears when the generation of ROS is more effective than their removal. The production of ROS occurs continuously in cells during catalysis by some enzymes (e.g., NADPH oxidase, xanthine oxidase, cyclooxygenase) and electron transport in the inner mitochondrial membrane. ROS at lower concentrations are “second messengers” and influence many cytophysiological processes. Moreover, cells regulate the level of ROS by a defense system that includes antioxidant enzymes and nonenzymatic antioxidants, e.g., a reduced form of glutathione [34,35]. The overproduction of ROS and oxidative stress induce oxidative damage of membrane lipids, enzymes, and structural proteins as well as nucleic acids [36]. ROS-mediated cellular damage may lead to cell cycle checkpoint arrest, inhibition of the cell cycle, or induction of apoptosis [37,38]. For this reason, the induction of oxidative stress is thought to be one of the mechanisms of anticancer action. Pro-oxidant therapies include radiotherapy as well as the treatment with anthracyclines, mitomycin C, bleomycin, procarbazine, cisplatin, and other drugs [39,40]. It is well known that stimulation of ROS production is also involved in drug-induced phototoxicity [41,42].

We demonstrated that both tested tetracyclines significantly increased the intracellular level of ROS in melanoma cells exposed to UVA radiation. The pro-oxidative effect caused by phototoxic action was much stronger for doxycycline than for chlortetracycline. A similar observation was made for glutathione. The level of reduced GSH was drastically decreased in the melanoma cells treated with tetracyclines and irradiated simultaneously. Of note, only minor redox imbalance was observed in cells exposed only to the drug or UVA. This indicates the use of glutathione to reduce ROS to less toxic molecules. GSH has a multidirectional antioxidant activity, including the detoxification of hydrogen peroxide and lipid peroxides via glutathione peroxidase. It also directly donates hydrogen or electrons to the reduction of other compounds. Thus, the GSH/GSSG ratio belongs to major determinants of oxidative stress [43,44]. With regard to the above, it can be stated that induction of oxidative stress is mainly and selectively related to phototoxic action.

Cell proliferation can also be regulated by ROS generated by mitochondria. These ATP-producing organelles also belong to major producers of endogenous ROS in cells [45]. Moreover, it was found that the progression of cell division and mitochondrial homeostasis were coordinated and interdependent. Mitochondrial ROS participate in the regulation of the redox balance and cell signaling pathways, e.g., MAPK/ERK or PI3K-AKT-mTOR [46,47]. It was also shown that cell cycle arrest was related to mitochondrial oxidative stress and reduced mitochondrial membrane potential [48]. The results presented in this article demonstrated that doxycycline or chlortetracycline alone, as well as in the combination with UVA radiation, depolarized the mitochondrial membrane in melanoma cells. Nevertheless, the highest values of the percentage of cells with a decreased mitochondrial membrane potential were observed in cells exposed to the tested drug and UVA simultaneously. Therefore, it can be concluded that one of the phototoxic effects of tetracyclines on melanoma cells is the depolarization of mitochondrial membranes. This action may lead not only to the deregulation of the cell cycle and redox imbalance, but may also contribute to cell apoptosis via the activation of the intrinsic pathway.

Changes in the polarity of the mitochondrial membranes and an increase in their permeability cause the release of pro-apoptotic factors, including cytochrome c and apoptosis-inducing factor (AIF) from mitochondria into the cytosol. Released cytochrome c participates in the formation of a protein complex known as the apoptosome which leads to the activation of the initiator caspase-9 [49]. In addition to the intrinsic pathway, there is an extrinsic pathway of apoptosis activation. In this case, initiation of apoptosis requires activation of transmembrane death receptors (members of the tumor necrosis factor receptor family, e.g., FasR and TNFR1) via specific ligands (e.g., FasL, TNF-α). After binding the ligand to the receptor, a death-inducing signaling complex (DISC) is formed and then the initiator caspase-8 is activated. In the next step, initiator caspases activate effector or “executioner” caspase-3/7 activity. The activation of the execution pathway triggers, among others, degradation of cytoskeletal and nuclear proteins as well as chromosomal DNA [50].

Regardless of the activation pathway, the characteristic feature of apoptosis is phosphatidylserine externalization, which is detected by labeled annexin V [51,52]. Cytometric analysis of annexin V indicated that doxycycline and chlortetracycline in the tested concentrations caused a slight increase in the percentage of annexin V-positive melanoma cells. In turn, a very high percentage of apoptotic cells was observed in cultures exposed simultaneously to the drug and UVA radiation. A slightly stronger effect was caused by the phototoxic effect of doxycycline which increased the percentage of apoptotic cells to around 90% in COLO 829 and G-361 cultures. The evaluation of caspases revealed that the combination of UVA and doxycycline highly activated initiator caspase-8 and -9 (over 90% of cells) as well as the effector caspase-3/7 (87% of COLO 829 cells and 96% of G-361 cells) in both cell lines. The effect of treatment with chlortetracycline and UVA radiation was weaker and more selective. The exposure of G-361 cells simultaneously to CHL and UVA caused the activation of all analyzed caspases. However, the activation of caspase-9 was not observed in COLO 829 cells after the treatment. It is also worth emphasizing that the combination of CHL and UVA triggered the activation of the effector caspase-3/7 only in 22% of COLO 829 cells and 44% of G-361 cells.

Differences between chlortetracycline and doxycycline, as well as between the tested cell cultures, were also revealed by DNA fragmentation analysis. A higher percentage of cells with fragmented DNA was observed in G-361 cells. The obtained results for combined therapy were around 80% for doxycycline and 40% for chlortetracycline. In turn, the significant increase in the percentage of COLO 829 cells with fragmented DNA was found only after simultaneous exposure to DOX and UVA and accounted for about 40%. Therefore, the degree of DNA fragmentation in the examined melanoma cells was correlated with the level of caspase-3/7 activity. Both analyses seem to be useful in the actual assessment of melanoma cell apoptosis. Together they combine the study of executive mechanisms of apoptosis and the effects of these mechanisms. The described correlation also suggests that DNA fragmentation may be mainly a result of cell apoptosis, and not its cause, evoked for example by free radical damage. The analysis of the final stages of apoptosis seems to be extremely important. It was found that melanoma cells can undergo partial apoptosis without dying. Failed apoptosis was observed in cells with partial mitochondrial permeabilization and non-lethal caspase activation. Moreover, melanoma cells surviving apoptosis gained migration and invasion properties [53]. Therefore, complete apoptosis and DNA damage by activation of caspases are the main goals of anticancer therapies [54,55].

Some studies have proved that combined treatment with irradiation, such as photodynamic therapy, might lead to DNA damage and induction of apoptosis in melanoma cells via induction of oxidative stress [56,57,58]. One of the cellular proteins activated by ROS and DNA damage is p53. This protein, known as a guardian of genomic stability, is a well-known transcription factor influencing the cell cycle, DNA repair, and apoptosis also in melanoma cells [59,60]. The regulation of gene expression by p53 in melanoma cells is related to the multidirectional action and participates in cell signaling, caspase activation, induction of apoptosis as well as chemosensitivity to therapy. For these reasons, p53 has become a target for potential therapeutics [61,62,63]. The presented analysis of confocal imaging indicated that p53 also contributed to the anti-melanoma effect related to the phototoxic action of tetracyclines. COLO 829 and G-361 cells exposed simultaneously to UVA radiation and DOX or CHL showed relatively high expression of the protein. A slightly increased level of p53 was also observed in G-361 melanoma cells treated with doxycycline and chlortetracycline. In other samples, no significant changes were found. Overall, the results are consistent with the analyses described above. It can be concluded that the increase in the p53 level is evidence of damage to melanotic melanoma cells by the phototoxic effect of tetracyclines.

Although the results presented above are promising, research on the use of tetracycline-induced phototoxicity in the treatment of melanoma must be continued. This study was based on the established experimental model, which allowed the comparison of the cytotoxic and phototoxic effects of doxycycline and chlortetracycline concerning the proliferation, apoptosis, and death of melanoma cells. A single drug application, a relatively low dosage of UVA radiation as well as short observation time may have contributed to the need to use high concentrations of the tested drugs. However, there is a high probability that changes in some experimental conditions will significantly reduce the effective concentrations of tetracyclines. The use of the drugs at lower concentrations may be possible also due to the high phototoxic potential and the accumulation in pigmented tissues.

## 4. Materials and Methods

### 4.1. Cell Culture and Treatment

Human melanotic melanoma cells COLO829 and G361, purchased from ATCC (CRL-1974, Manassas, VA, USA), were incubated in a humidified 5% CO_2_ incubator at 37 °C. The cells were cultured in the recommended medium containing L-glutamine and supplemented with inactivated fetal bovine serum and the following antibiotics: penicillin (100 U/mL), neomycin (10 µg/mL) and amphotericin B (0.25 µg/mL). COLO829 cells were maintained in Roswell Park Memorial Institute (RPMI) 1640 medium (PAN-Biotech, Aidenbach, Germany), and G361 cells were cultivated in McCoy’s 5A Medium (Sigma Aldrich Inc., St. Louis, MO, USA).

Treatment with chlortetracycline and doxycycline was started 24 h after seeding for melanoma cells. The initial stock solutions of the drugs were made by dissolving the substances in water. The tetracycline solutions were obtained by diluting the stock solutions in a culture medium appropriate for the cell line. The screening analysis using the WST-1 assay was made for the drug concentrations ranging from 10 µM to 200 µM. Other studies were performed for doxycycline at a concentration of 50 µM and for chlortetracycline at a concentration of 150 µM. After 24-h treatment of melanoma cells, the drug solutions were replaced with phosphate buffered saline (PBS) (Thermo Fisher Scientific, Waltham, MA, USA) and then the cells were irradiated with UVA at a dose of 2 J/cm^2^ using a filtered lamp BVL-8.LM (Vilber Lourmat, France). After irradiation, the cells were incubated in a basic culture medium until analysis.

### 4.2. Cell Viability Assessment

Cell viability was assessed using the NucleoCounter^®^ NC-3000™ fluorescence image cytometer (ChemoMetec, Lillerød, Denmark) according to the Cell Viability and Cell Count Assays protocol. In brief, after the trypsinization, the cells were centrifuged and suspended in the growth medium and then loaded into the Via1-Cassette™ containing two fluorescent dyes: DAPI and acridine orange (ChemoMetec, Lillerød, Denmark). The immediate analysis causes DAPI penetration only through a damaged and permeable cell membrane of dead cells. In turn, acridine orange allows the detection of the total cell population.

### 4.3. WST-1 Assay

WST-1 reagent (Roche GmbH, Mannheim, Germany) is a tetrazolium salt, which is cleaved to formazan by mitochondrial dehydrogenases. The assay principle assumes that the amount of formazan day is proportional to the number of metabolically active cells. In brief, the tested melanoma cells were seeded in 96-well microplates and treated, as described above. Afterwards, the WST-1 (10 µL/well) was added to the cells cultured with 100 µL of the growth medium/well. Finally, the cells were incubated for the next 3 h at 37 °C and 5% CO_2_. The absorbance of samples were measured at 440 nm and 650 nm as a reference wavelength. Microplate reader Infinite 200 PRO (TECAN, Männedorf, Switzerland) was used for the measurements.

### 4.4. The Quantitation of Cellular Reactive Oxygen Species

CellROX Green Reagent (Thermo Fisher Scientific, Waltham, MA, USA) was used to quantify ROS in the analyzed melanoma cells. This cell-permeable reagent has a strong fluorogenic signal upon oxidation by ROS in live cells with absorption/emission maxima of about 485/520 nm. Briefly, the cells were cultured in 96-well black, nontransparent microplates. After the treatment, the cells were incubated with 5 µM solution of CellROX Green Reagent for 30 min and then washed with PBS. Fluorescence measurements (excitation/emission: 485 nm/520 nm) were made using the microplate reader Infinite 200 PRO. The results were normalized to the number of metabolically active cells.

### 4.5. Assessment of the Intracellular Thiol Status

The intracellular level of thiols was evaluated cytometrically using VitaBright 48 (ChemoMetec, Lillerød, Denmark) staining. The reagent allows assignment of the cell population with a high level of reduced thiols, such as GSH. The analysis was made according to the Vitality Assay protocol using the NucleoCounter^®^ NC-3000™ imaging cytometer. After the treatment, the investigated melanoma cells were suspended in PBS (2 × 10^6^ cells/mL) and stained with VitaBright-48™ (10 µL/190 µL of the cell suspension). Finally, the analysis was made according to the Vitality Assay protocol using the NC-3000 image cytometer.

### 4.6. Annexin V Assay

Annexin V Assay allows assessment of apoptosis in the tested cell populations. The assay principle is based on a high affinity of annexin V to phosphatidylserine, which is externalized in apoptotic cells. After the treatment, about 3.0 × 10^5^ cells were suspended in 100 µL of Annexin V Binding Buffer (ChemoMetec, Lillerød, Denmark) with 2 µL of Hoechst 33,342 (ChemoMetec, Lillerød, Denmark) and 2 µL of FITC-labeled annexin V (Biotium, Fremont, CA, USA). Afterwards, the cells were incubated for 15 min at 37 °C and subsequently centrifuged. The obtained cell pellets were washed twice using Annexin V Binding Buffer. Finally, the cell pellets were resuspended in 100 µL of Annexin V Binding Buffer with 2 µL of propidium iodide (ChemoMetec, Lillerød, Denmark) staining late apoptotic and necrotic cells. The analysis was made using the NucleoCounter^®^ NC-3000™ fluorescence image cytometer.

### 4.7. Caspase Activity Assay

The activity of caspase-3/7, 8, 9 was determined cytometrically using assays based on fluorochrome-labeled inhibitor of caspases—FLICA reagents: Caspase 9 Assay Kit, Caspase 8 Assay Kit and Caspase 3/7 Assay Kit (ImmunoChemistry Technologies, Bloomington, MN, USA). After the treatment, melanoma cells were detached, counted and suspended in PBS (4 × 10^6^ cells/mL). Next, 5 μL of FLICA reagent and 2 μL of Hoechst 33,342 was added to 93 µL of cell suspensions and incubated for 60 min at 37 °C. Then, the samples were washed twice with 400 µL of Apoptosis Wash Buffer (ChemoMetec, Lillerød, Denmark), centrifuged, and finally resuspended in 100 µL of Apoptosis Wash Buffer supplemented with propidium iodide (10 μg/mL). The analysis was made according to the Caspase Assay protocol using the NucleoCounter^®^ NC-3000™.

### 4.8. DNA Fragmentation Assay

The DNA fragmentation analysis of melanoma cells was performed using the fluorescence image cytometer NucleoCounter^®^ NC-3000™. After the treatment, the cells were detached, counted and fixed with 70% cold ethanol (24 h at 0–4 °C). Afterwards, the samples were washed with PBS, centrifuged and stained with Solution 3 (ChemoMetec, Lillerød, Denmark) containing DAPI and 0.1% triton X-100 in PBS. The analysis was made according to the DNA Fragmentation Assay protocol.

### 4.9. Cell Cycle Analysis

The analysis of the cell cycle in the tested melanoma cells was performed using the fluorescence imaging cytometer NucleoCounter^®^ NC-3000™. In brief, after the treatment, the cells were detached, counted and fixed with 70% cold ethanol for at least 24 h at 0–4 °C. After fixation, the samples were centrifuged and then the ethanol was removed. The obtained cell pellets were washed with PBS and finally resuspended in the staining Solution 3. The tested cells were analyzed according to the Fixed Cell Cycle-DAPI Assay protocol.

### 4.10. Mitochondrial Potential Analysis

The potential of the mitochondrial membrane was evaluated using the image cytometry technique, which allows demarcation of the percentage of cells with a polarized or depolarized mitochondrial membrane. The analysis is based on a fluorescent cationic dye JC-1 accumulating inside the mitochondria with a high transmembrane potential. At high concentrations, JC-1 aggregates and shows red fluorescence. In turn, JC-1 localizes in the cytoplasm in cells with a low mitochondrial potential and then shows green fluorescence. The tested melanoma cells were stained with JC-1 after the treatment by adding 12.5 µL of Solution 7 (200 µg/mL JC-1) (ChemoMetec, Lillerød, Denmark) to cell suspensions in PBS (1.0 × 10^6^ cells/mL). Then, the samples were incubated for 30 min at 37 °C, centrifuged and washed twice with PBS. Immediately before the analysis, the cells were suspended with 0.25 mL of Solution 8 (1 µg/mL DAPI in PBS) (ChemoMetec, Lillerød, Denmark). The analysis was made according to the Mitochondrial Potential Assay protocol using NucleoCounter^®^ NC-3000™.

### 4.11. Immunofluorescence Analysis by Confocal Microscopy

Laser confocal microscope Nikon Eclipse Ti-E A1R-Si and Nikon NIS Elements AR software were used to cell visualization. Melanoma cells were cultured on sterile coverslips in Petri dishes in the recommended growth media. After the treatment procedure, the cells were fixed and permeabilized with 4% paraformaldehyde and 0.1% Triton X-100, respectively. Before the staining, samples were incubated with glycine and bovine serum albumin. Afterwards, the cells were incubated with primary rabbit anti-p53 antibody (1:1000) (Cell Signaling, Danvers, MA, USA) at 4 °C for 24 h. Then, the samples were stained with Alexa Fluor 488-conjugated anti-rabbit secondary antibody (1:200) (Thermo Fisher Scientific, Waltham, MA, USA). The labeling of cell nuclei with SYTO Deep Red Nucleic Acid Stain (Thermo Fisher Scientific, Waltham, MA, USA) as well as labeling of actin filaments with Phalloidin–Atto 565 (Sigma-Aldrich Inc., Taufkirchen, Germany) were also applied according to the manufacturer’s protocol.

### 4.12. Statistical Analysis

Statistical analysis was performed using GraphPad Prism 6.01 software. The calculations included mean values of at least three separate experiments performed in triplicate (n = 9) ± standard deviation of the mean (SD). In addition, statistical significance was determined by one-way ANOVA and two-way ANOVA, as well as Dunnett’s and Tukey’s multiple comparison tests. The Kolmogorov–Smirnov test and the Brown–Forsythe test were used to check the compliance of the distribution results as well as the homogeneity assumption of variances, respectively. A *p* value of less than 0.05 was statistically significant.

## 5. Conclusions

In conclusion, we have demonstrated for the first time that the phototoxic action of tetracyclines can be used against melanotic melanoma. The analysis indicated that the combined therapy caused a significant decrease in the number of live cells in the tested cultures. In general, the effect was a result of cell cycle arrest as well as lowered cell viability. It was found that treated cells had depolarized mitochondria, a low level of reduced thiols, and a high level of ROS. Moreover, the study showed that simultaneous exposure of cells to drugs and UVA radiation induced apoptosis. The programmed cell death was confirmed by positive results of annexin V assay and activation of caspases. Finally, DNA fragmentation and the increased level of p53 were observed in melanoma cells as a result of the phototoxic action of tetracyclines. In general, the study revealed that doxycycline demonstrated a stronger cytotoxic and phototoxic effect. The possibility of the local application of drugs in skin cancers and precise irradiation makes it possible to use the phototoxicity of tetracyclines in the treatment of melanotic melanoma. Although the tetracyclines themselves have anticancer effects, including anti-melanoma, the additional use of UVA radiation allows for the reduction of therapeutic concentrations and increases the effectiveness of the tested drugs.

## Figures and Tables

**Figure 1 ijms-24-02353-f001:**
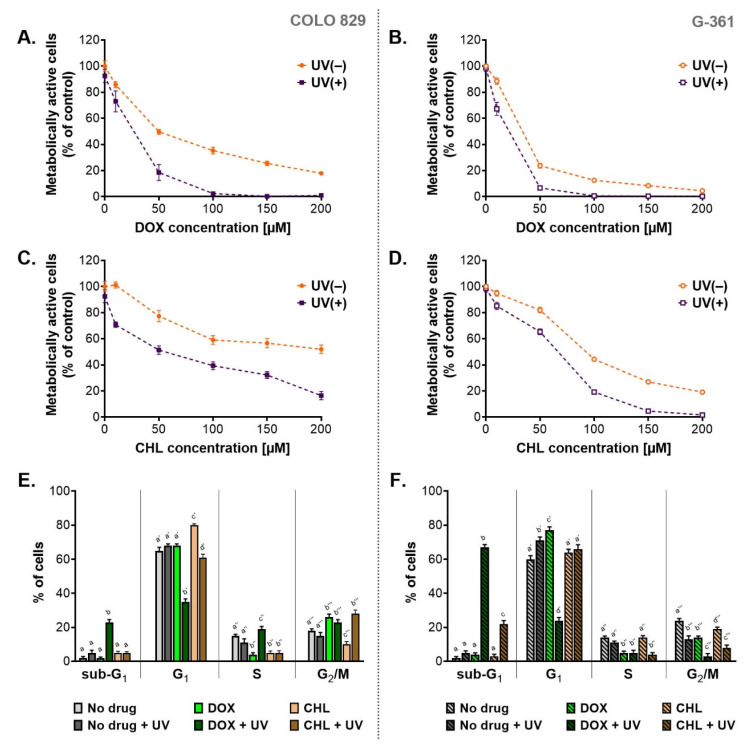
The assessment of cytotoxic and phototoxic effect of doxycycline and chlortetracycline on melanotic melanoma cell lines. The analysis of the number of metabolically active cells based on the WST-1 assay (**A**–**D**). Results of the cytometric analysis of the cell cycle (**E**,**F**). All graphs present mean values ± SD. Using different letters to show statistical significance.

**Figure 2 ijms-24-02353-f002:**
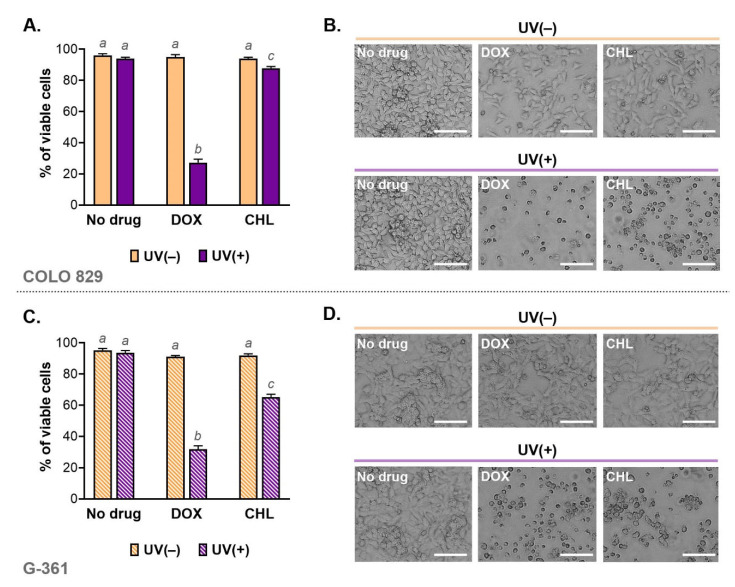
Viability of melanoma cells exposed to doxycycline and chlortetracycline as well as UVA radiation (**A**,**C**). The results were expressed as mean values ± SD. Sample microscopic images showing morphology of the tested melanoma cell cultures after the investigated treatment (**B**,**D**). Scale bar 250 µm. Using different letters to show statistical significance.

**Figure 3 ijms-24-02353-f003:**
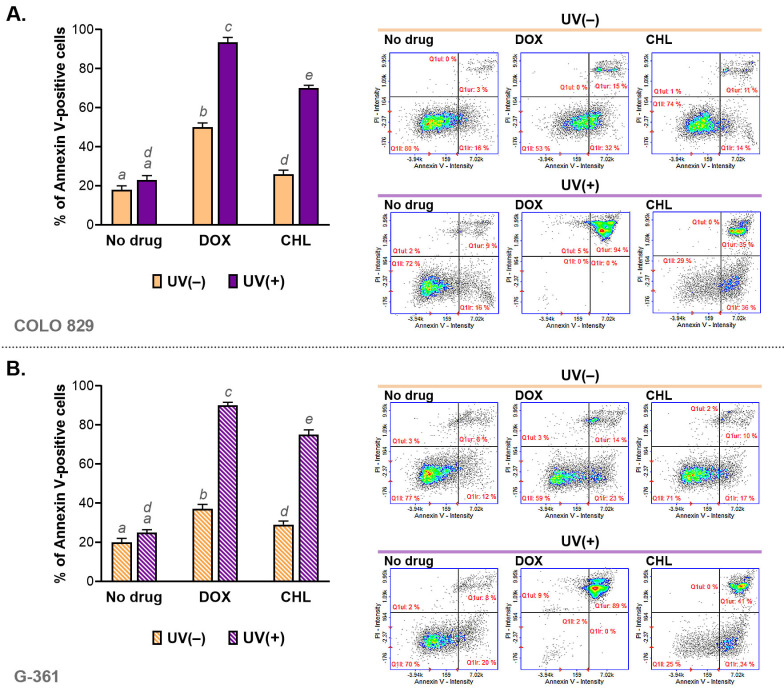
The cytometric analysis of annexin V in melanoma cells COLO 829 (**A**) and G-361 (**B**) incubated with doxycycline and chlortetracycline and irradiated with UVA. Bar graphs present mean values ± SD. Included scatter plots show representative results for the analysis of the tested cell populations. Using different letters to show statistical significance.

**Figure 4 ijms-24-02353-f004:**
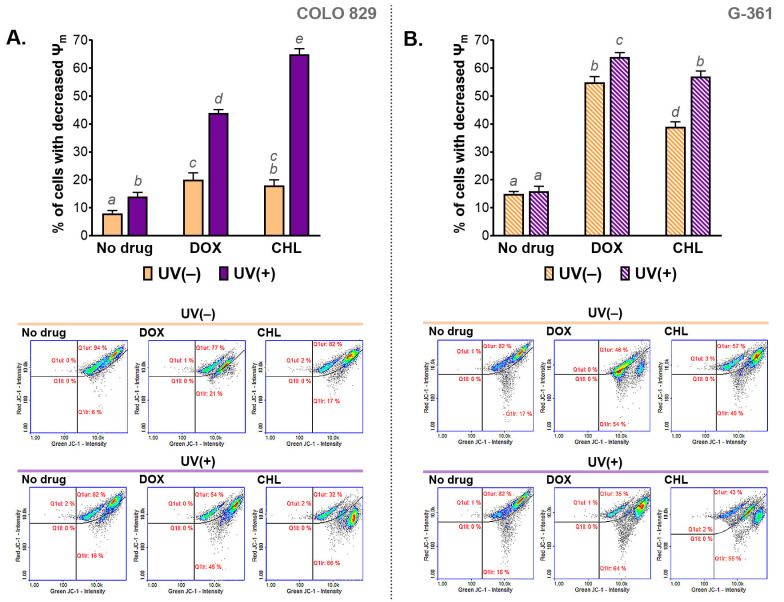
The analysis of the mitochondrial membrane potential (ψm) in melanoma cells COLO 829 (**A**) and G-361 (**B**) incubated with doxycycline and chlortetracycline and exposed to UVA radiation. Mean values ± SD were presented in bar graphs. Sample scatter plots show representative results for the analysis of the tested cell populations. Using different letters to show statistical significance.

**Figure 5 ijms-24-02353-f005:**
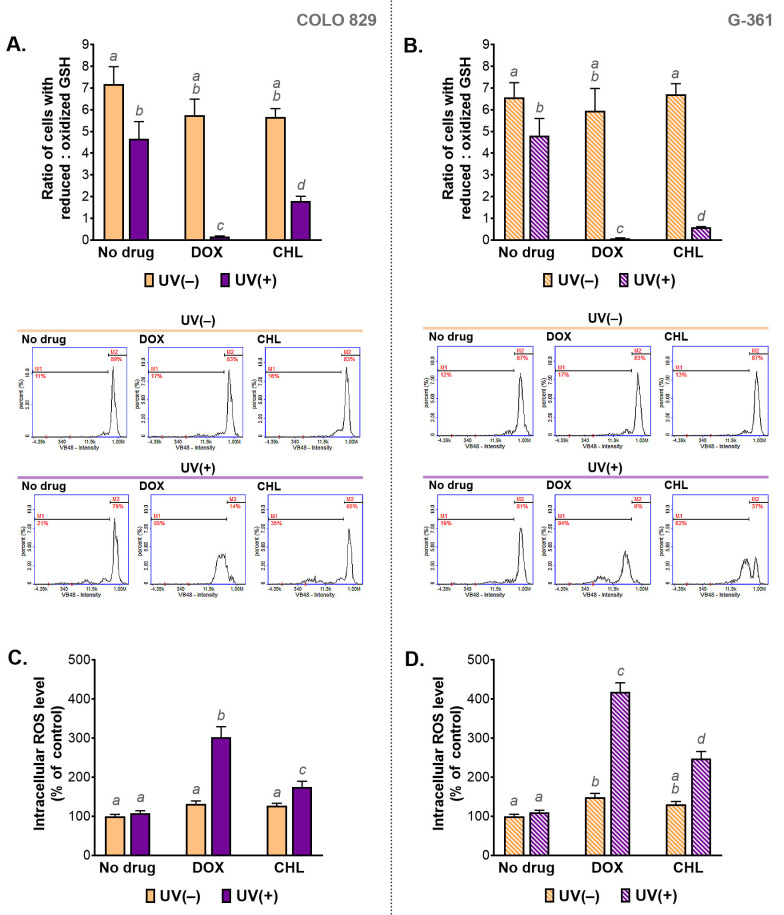
The analysis of the intracellular level of thiols (**A**,**B**) and reactive oxygen species (**C**,**D**) in melanoma cells treated with doxycycline and chlortetracycline and irradiated with UVA. Bar graphs present mean values ± SD. Included histograms are representative results of cytometric analysis of thiols. Using different letters to show statistical significance.

**Figure 6 ijms-24-02353-f006:**
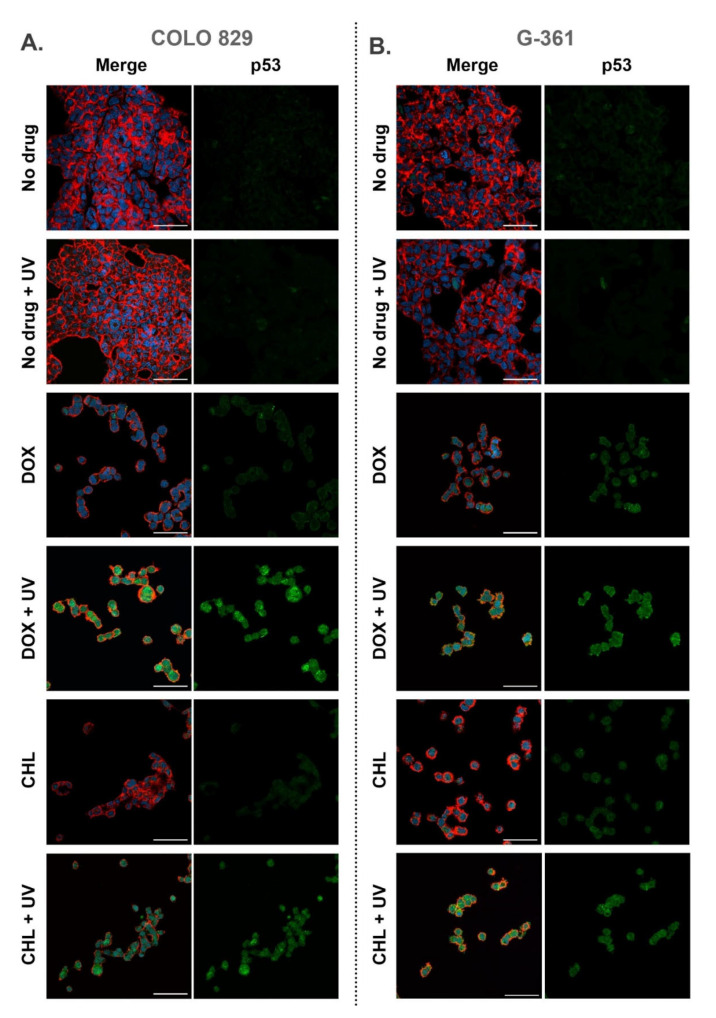
Confocal imaging of COLO 829 (**A**) and G-361 (**B**) melanotic melanoma cells treated with doxycycline and chlortetracycline and exposed to UVA radiation. The figure present merged images (nuclei—blue, actin cytoskeleton—red, and p53—green) as well as the separate channel visualizing p53. Scale bar 50 µm.

**Figure 7 ijms-24-02353-f007:**
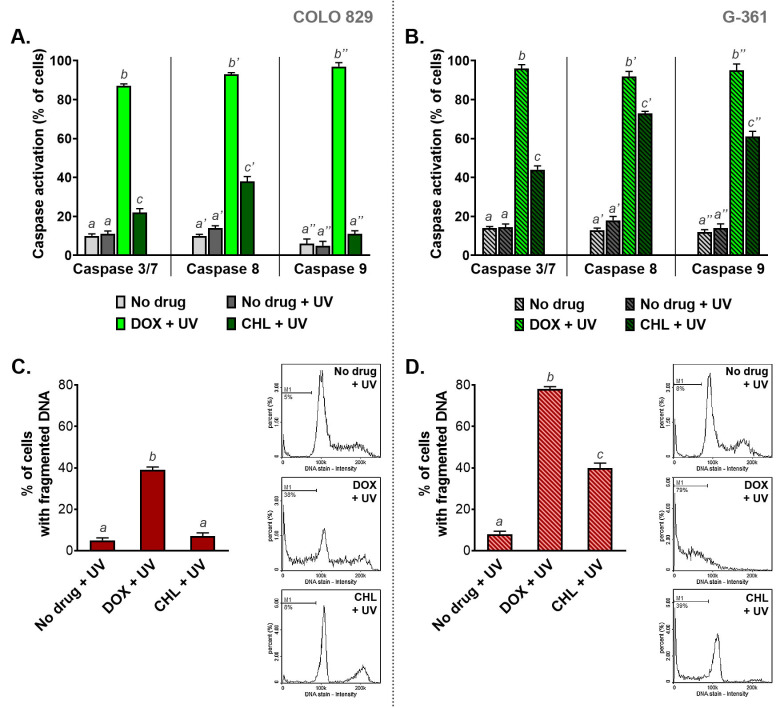
The analysis of caspase activation (**A**,**B**) and DNA fragmentation (**C**,**D**) in melanoma cells treated with doxycycline and chlortetracycline and irradiated with UVA. Bar graphs present mean values ± SD. Included histograms are representative results of cytometric analysis of DNA fragmentation. Using different letters to show statistical significance.

**Table 1 ijms-24-02353-t001:** Calculated values of EC_50_ and PIF for doxycycline and chlortetracycline after the treatment of melanotic melanoma cell lines COLO 829 and G-361.

Cell Line	Doxycycline			Chlortetracycline		
	UV(−)	UV(+)	PIF*	UV(−)	UV(+)	PIF *
**COLO 829**	51.1 µM	15.0 µM	3.4	185.2 µM	52.0 µM	3.6
**G-361**	23.3 µM	11.5 µM	2.0	85.5 µM	38.6 µM	2.2

* Photo-irritation factor (PIF) = EC_50_UV(−)/EC_50_UV(+).

**Table 2 ijms-24-02353-t002:** The analysis of the cell cycle presented as a relative ratio of G_1_/S and G_2_-M/S for COLO 829 and G-361 melanoma cells treated with DOX and CHL and exposed to UVA radiation.

Cell Line	Coefficient	Control	UV(+)	DOX UV(−)	DOX UV(+)	CHL UV(−)	CHL UV(+)
**COLO 829**	**G_1_/S**	4.3	6.2	17.0	1.8	16	12.2
	**G_2_-M/S**	1.2	1.5	6.5	1.2	2.0	5.6
**G-361**	**G_1_/S**	4.3	6.5	15.4	4.0	4.6	16.5
	**G_2_-M/S**	1.7	1.2	2.8	0.5	1.4	2.0

## Data Availability

The data presented in this study are available upon request from the corresponding author.

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
