# Peer review of "The Assessment of the Phototoxic Action of Chlortetracycline and Doxycycline as a Potential Treatment of Melanotic Melanoma—Biochemical and Molecular Studies on COLO 829 and G-361 Cell Lines"

_ijms, 2023, doi:10.3390/ijms24032353_

Round 1

Reviewer 1 Report

I would like to thank the Academic Editor for giving me the opportunity to review this interesting manuscript on the use of antimicrobial chemotherapeutics such as tetracyclines in the use towards such an aggressive neoplasm as melanotic melanoma, exploiting their phototoxicity and cytotoxicity properties. The authors conduct an analysis on two MM cell lines and produce results showing that these molecules are able to slow down and interfere with the cell cycle. I believe, overall, that after an English language check, these results can be published.

Author Response

Dear Reviewer,

We would like to thank you for the proofreading and the assessment of our article as well as for your kind opinion. At the same time, we apologize for language errors that have been found in the work. In response to your suggestion, the corrections have been introduced to the manuscript (edited fragments are marked in yellow). We hope the improved version of the paper will be satisfying.

Reviewer 2 Report

Lines 37-39: “Both drugs and other types of therapies have failed  to provide adequate therapeutic effects, i.e. inhibition of tumor growth and significant  prolongation of the lives of patients with advanced metastases.” This statement is factually wrong. In the last decade at least four medicinal products with different mechanisms of activity have been authorized in melanoma in different regions of the world, particularly prolonging lives of patients with melanoma in the advanced stage. Whereas the new therapies do not yet cure melanoma, it is wrong to claim that the current therapies do not significantly prolong life. Rather, the authors should state that despite the progress made in recent years, which manage to extend the life of patients with metastatic disease, there is still place for improvement.

“In turn, chlortetracycline is the first-generation tetracycline.” This suggests that a single first-generation tetracycline exists, which is chlortetracycline. It should be phrased along the line of “, chlortetracycline is a first-generation tetracycline”.

Line 400: “inactivated fetal bovine serum and antibiotics”. Please nominate antibiotics used (and concentrations).

Lines 444-445: “The treatment with chlortetracycline and doxycycline was started 24 h after seeding  for melanoma cells. Then, the drug solutions were replaced with PBS….”. Please clarify solvents used, concentrations etc. The current description of the tetracycline treatment is very vague. The same for lines 492-493 (“In the  first step, the cells were treated with the studied drugs for 24 h” – please provide details on the treatment – solvent, concentrations etc). Figure 1 indicates that only concentrations in the micromolar range were used; these are quite high. Compounds of potential clinical relevance are those active in the nanomolar (or even picomolar ranges). The authors should justify the concentrations used (which, in our view, are too high to be clinically relevant). Moreover, the same figure suggests that even with UVA,  cytotoxic effects are obvious only for levels higher than 10-20 uM (not the very low micromolar range).

Section 4.2: please clarify source of reagents used (DAPI, acridine orange), concentrations etc, so as to allow an independent researcher to replicate the work.

Lines 477-480: were those stains used simultaneously or successively, after what time interval and in what concentrations?

Line 488: please describe or reference “apoptosis wash buffer”.

Lines 541-543: “The possibility of  local application of drugs in skin cancers and precise irradiation makes it possible to use  the phototoxicity of tetracyclines in the treatment of melanotic melanoma.” The preferred treatment is usually surgical excision (when the tumor is localized, i.e. before metastasizing); topical treatment is rather used when surgery is not possible or not accepted by the patient.

The authors do not discuss the limitations of the study (among which, in our view, the most important is the very high concentrations used in this study).

Author Response

Dear Reviewer,

We are grateful for a thorough and insightful assessment of our manuscript. The provided comments and advice are accurate and valuable. Having taken into consideration the remarks, we have carefully revised and corrected the manuscript. All edited fragments of the manuscript have been marked in yellow. We hope that the improved version of the paper will be satisfying for both the Reviewer and the Editor.

Our work has been linguistically revised. Moreover, the Materials and Methods section has been extensively changed and rewritten in more detail considering the Reviewer comments.  

Reviewer’s comments: Lines 37-39: “Both drugs and other types of therapies have failed  to provide adequate therapeutic effects, i.e. inhibition of tumor growth and significant  prolongation of the lives of patients with advanced metastases.” This statement is factually wrong. In the last decade at least four medicinal products with different mechanisms of activity have been authorized in melanoma in different regions of the world, particularly prolonging lives of patients with melanoma in the advanced stage. Whereas the new therapies do not yet cure melanoma, it is wrong to claim that the current therapies do not significantly prolong life. Rather, the authors should state that despite the progress made in recent years, which manage to extend the life of patients with metastatic disease, there is still place for improvement.

Response: We would like to thank you for the suggestion. We fully agree with the recommendation. In response, we changed this fragments as follows:

“Currently applied therapeutic methods of melanoma skin cancer include, among others, surgical resection, chemotherapy and photodynamic therapy. In addition, targeted therapy and immunotherapy have been introduced in recent years. Although these methods have resulted in the extension of patients’ lives, new and better therapies with better safety and higher clinical effectiveness are still being sought [8, 9].”

  1. Domingues, B.; Lopes, J.M.; Soares, P.; Pópulo, H. Melanoma treatment in review. Immunotargets Ther. 2018, 7, 35-49. doi: 10.2147/ITT.S134842.
  2. Lazaroff, J.; Bolotin, D. Targeted therapy and immunotherapy in melanoma. Dermatol. Clin. 2023, 41, 65-77. doi: 10.1016/j.det.2022.07.007.

Reviewer’s comments: Figure 1 indicates that only concentrations in the micromolar range were used; these are quite high. Compounds of potential clinical relevance are those active in the nanomolar (or even picomolar ranges). The authors should justify the concentrations used (which, in our view, are too high to be clinically relevant). Moreover, the same figure suggests that even with UVA,  cytotoxic effects are obvious only for levels higher than 10-20 µM (not the very low micromolar range).

Response: The remark concerning drug concentrations is very valid, especially referring to plasma therapeutic concentrations of tetracyclines. The choice of initial range of drug concentrations was based on previous research on melanoma cells and normal melanocytes. Thus, we decided to make screening analysis using the drugs in concentrations from 10 µM to 200 µM. The results of WST-1 assay indicated the therapeutic potential and efficacy occurred in this range. In turn, the results of screening analysis were used to select concentrations for further studies. It is worth emphasizing that calculated values for PIF as well cytometric studies and microscopic observations have revealed the intensification of tetracycline cytotoxicity to melanoma cells after the exposure to UVA . It should be also added that the applied dosage of UVA radiation is relatively low. The exposure parameters are analogous to previous experiments on normal melanocytes to compare phototoxic effects. It is very probable that the prolongation of irradiation time would intensify the cytotoxic effects of the drugs and would let to reduce the effective tetracycline concentrations. Our study presents a short-term, single application experimental model to show the general potential and mechanisms of such treatment. Further research will certainly include an attempt to develop new experimental models that would contribute to reducing the used concentrations. It should take into account, among others, the possibility of developing a modern drug delivery form as well the possibility of tetracycline accumulation in pigmented tissues.

Reviewer’s comments: Lines 541-543: “The possibility of  local application of drugs in skin cancers and precise irradiation makes it possible to use the phototoxicity of tetracyclines in the treatment of melanotic melanoma.” The preferred treatment is usually surgical excision (when the tumor is localized, i.e. before metastasizing); topical treatment is rather used when surgery is not possible or not accepted by the patient.

Response: We fully share the opinion of the Reviewer. However, in our view, the statement indicates only the possibility of local treatment and does not recommend it as a first-line therapy. Research on the use of phototoxic properties of tetracyclines may contribute to the development of alternative therapeutic methods of melanoma. Local treatments, despite some clinical limitations, have several significant advantages, including minimal invasive character and selectivity as well relatively low side effects. 

Reviewer’s comments: The authors do not discuss the limitations of the study (among which, in our view, the most important is the very high concentrations used in this study).

Response: Thank you for the valuable remark. With reference to the comment, a new fragment has been added to the manuscript.

“Although the results presented above are promising, research on the use of tetracycline-induced phototoxicity in the treatment of melanoma must be continued. This study was based on the established experimental model, which allowed to compare the cytotoxic and phototoxic effects of doxycycline and chlortetracycline concerning the proliferation, apoptosis, and death of melanoma cells. A single drug application, a relatively low dosage of UVA radiation as well as short observation time may have contributed to the need to use high concentrations of the tested drugs. However, there is a high probability that changes in some of experimental conditions will significantly reduce the effective concentrations of tetracyclines. The use of the drugs at lower concentrations may be possible also due to the high phototoxic potential and the accumulation in pigmented tissues.”

Reviewer 3 Report

Dear Authors,

Thank you for your interesting contribution. In future research, I recommend including a non-cancerous cell line as a control.

However, I would like some clarification of the information found in lines 143-146 and Figure 2.

Lines 143-146: “Cell viability was analyzed using imaging cytometry. The obtained results for individual samples and the corresponding microscopic images were shown in Figure 2. Based on the performing studies, it was stated that the viability of cells exposed only to doxycycline in a concentration of 50 µM or chlortetracycline in a concentration of 150 µM was high and not different from controls.”

If I understand correctly, doxycycline in a concentration of 50 µM caused no decrease in the cellular viability, which is rather strange since, in Table 1, concentrations of 51.1 µM and 23.3 µM were presented as EC50 values towards COLO 829 and  G-361 cells. Also, Figure 2A shows almost 100% viability of DOX/UV-, whereas Figure 2B shows that number of cells in DOX/UV- is visibly lower than untreated cells. The same observations can be seen for chlortetracycline-treated cells. Please explain.

Author Response

Dear Reviewer,

First of all, we would like to thank you for the assessment of our manuscript and your favorable opinion.

We fully agree with the recommendation concerning the studies with the non-cancerous cell line. Concurrently, we would like to mention that normal epidermal melanocytes have been used in our previous studies of the phototoxic potential of doxycycline and chlortetracycline (*Rok, J.; Buszman, E.; Beberok, A.; Delijewski, M.; Otręba, M.; Wrześniok, D. Modulation of melanogenesis and antioxidant status of melanocytes in response to phototoxic action of doxycycline. Photochem. Photobiol. 2015, 91, 1429–1434. doi: 10.1111/php.12497; *Rok, J.; Rzepka, Z.; Respondek, M.; Beberok, A.; Wrześniok, D. Chlortetracycline and melanin biopolymer—The risk of accumulation and implications for phototoxicity: An in vitro study on normal human melanocytes. Chem. Biol. Interact. 2019, 303, 27–34. doi: 10.1016/j.cbi.2019.02.005). Both articles were discussed in the paper.

The EC50 values were calculated based on the WST-1 screening analysis. The test uses enzymatic cleavage of a tetrazolium salt to formazan dye. The reaction is catalyzed by mitochondrial dehydrogenases. Thus, it is assumed that the amount of formazan in a tested sample is proportional to the number of metabolically active cells. This principle makes that the final result can be the effect of the anti-proliferative action, the induction of cell death, or both of them. The assay does not indicate directly the reason for the observed decrease in the number of living and metabolically active cells. The relatively low specificity of the test causes additional analyses to be necessary to find the answer to the question about the mechanisms. Therefore, we conducted among others studies of cell viability and cell cycle. In addition, we made photographic documentation of microscopic analyses. In our opinion, final conclusions can be drawn only based on multi-parameter analyses. In the case of DOX/UV(-) and CHL/UV(-), inhibition of cell proliferation seems to be the main effect causing low values in WST-1 analysis. To summarize, we use the WST-1 test for screening analysis as well as to initial assessment. It is worth emphasizing that WST-1 is a useful assay to compare the general effectiveness of different treatments as well as to select drug concentrations for more detailed research. 

We hope the explanations will be satisfying and sufficiently clear for the Reviewer.